# On the Positive Definiteness of the Neural Tangent Kernel

## Abstract

The Neural Tangent Kernel (NTK) has emerged as a fundamental concept in the study of wide Neural Networks. In particular, it is known that the positivity of the NTK is directly related to the memorization capacity of sufficiently wide networks, i.e., to the possibility of reaching zero loss in training, via gradient descent. Here we will improve on previous works and obtain a sharp result concerning the positivity of the NTK of feedforward networks of any depth. More precisely, we will show that, for any non-polynomial activation function, the NTK is strictly positive definite.

## 1 Introduction

Recently, the increase in the size of deep neural networks (DNNs), both in the number of trainable parameters and the amount of training data resources, has been in step with the astonishing success of using DNNs in practical applications. This motivates the theoretical study of wide DNNs. In such context, the Neural Tangent Kernel (NTK) Jacot et al. (2018) as emerged as a fundamental concept. In particular, it is known that the ability of sufficiently wide neural networks to memorize a given training data set is related to the positivity of the NTK. More precisely, if the NTK is strictly positive definite then the quadratic loss will converge to zero, in the training via gradient descent, of an appropriately initialized and sufficiently wide feed-forward network (see for instance Du et al. (2018); Carvalho et al. (2023), and references therein). The positivity of the NTK has also been related to the generalization performance of DNNs (Cao & Gu, 2019; Arora et al., 2019; Chen et al., 2020).

Consequently, understating which conditions lead to this positivity becomes a fundamental problem in machine learning, and several works, that we will review later, have tackled this question providing relevant and interesting partial results. However, all of these results require nontrivial extra assumptions, either at the level of the training set, for instance by assuming that the data lies in the unit sphere, or at the level of the architecture, for instance by using a specific activation function, or both. The goal of the current paper is to obtain a sharp result in the context of feedforward networks which requires no such extra assumptions. In fact, we will show that for any depth and any non-polynomial activation function the corresponding (infinite width limit) NTK is strictly positive definite (see Section 1.2).

Finally, the proofs we present here are self-contained and partially based on an interesting characterization of polynomial functions (see Appendix A) which we were unable to locate in the literature and believe to have mathematical value in itself.

### 1.1 Feedforward Neural Networks and the Neural tangent kernel

Given $L \in \mathbb{Z}^+$, define a feedforward neural network with $L - 1$ hidden layers to be the function $f_\theta = f_\theta^{(L)} : \mathbb{R}^{n_0} \to \mathbb{R}^{n_L}$ defined recursively by the relations

$$f_\theta^{(1)}(x) = \frac{1}{\sqrt{n_0}} W^{(0)} x + \beta b^{(0)} , \tag{1.1}$$

$$f_\theta^{(\ell+1)}(x) = \frac{1}{\sqrt{n_\ell}} W^{(\ell)} \sigma(f_\theta^{(\ell)}(x)) + \beta b^{(\ell)} , \tag{1.2}$$

where the networks parameters $\theta$ correspond to the collection of all weight matrices $W^{(\ell)} \in \mathbb{R}^{n_{\ell+1} \times n_\ell}$ and bias vectors $b^{(\ell)} \in \mathbb{R}^{n_{\ell+1}}$, $\sigma : \mathbb{R} \to \mathbb{R}$ is an activation function, that operates entrywise when applied to vectors, and $\beta \geq 0$ is a fixed/non-learnable parameter used to control the intensity of the bias.

We will assume that our networks are initiated with *iid* parameters satisfying:

$$W \sim \mathcal{N}(0, \rho_W^2) \quad \text{and} \quad b \sim \mathcal{N}(0, \rho_b^2) \,, \tag{1.3}$$

with non-vanishing variances $\rho_W$ and $\rho_b$ .

For $\mu = 1, \ldots, n_L$ let $f_{\theta,\mu}^{(L)}$ the $\mu$-component of the output function $f_\theta^{(L)}$. It is well known (Radford, 1994; Lee et al., 2018) from the central limit theorem that, in the (sequential) limit $n_1, \ldots, n_{L-1} \to \infty$, i.e. when the number of all hidden neurons goes to infinity, the $n_L$ components of the output function $f_\mu^{(L)} : \mathbb{R}^{n_0} \to \mathbb{R}$ converge in law to independent centered Gaussian processes $f_{\infty,\mu}^{(L)} : \mathbb{R}^{n_0} \to \mathbb{R}$ with covariance $\hat{\Sigma}^{(L)} : \mathbb{R}^{n_0} \times \mathbb{R}^{n_0} \to \mathbb{R}$ , defined recursively by (compare with Yang (2019)):

$$\hat{\Sigma}^{(1)}(x, y) = \frac{\rho_W^2}{\sqrt{n_0}} x^\intercal y + \beta \rho_b^2 \,, \tag{1.4}$$

$$\hat{\Sigma}^{(\ell+1)}(x, y) = \rho_W^2 \, \mathbb{E}_{f \sim \hat{\Sigma}^{(\ell)}} \left[ \sigma(f(x)) \sigma(f(y)) \right] + \rho_b^2 \beta^2 \,. \tag{1.5}$$

A centered Gaussian Process $f$ with covariance $\Sigma$ will be denoted by $f \sim \Sigma$. Thus, for any $\mu \in \{1, \ldots, n_L\}$ we have $f_{\infty,\mu}^{(L)} \sim \hat{\Sigma}^{(L)}$. In particular, for $x, y \in \mathbb{R}^{n_0}$,

$$\left[ \begin{array}{c} f_{\infty,\mu}^{(L)}(x) \\ f_{\infty,\mu}^{(L)}(y) \end{array} \right] \sim \mathcal{N}\left( \left[ \begin{array}{c} 0 \\ 0 \end{array} \right], \left[ \begin{array}{cc} \hat{\Sigma}^{(L)}(x, x) & \hat{\Sigma}^{(L)}(x, y) \\ \hat{\Sigma}^{(L)}(y, x) & \hat{\Sigma}^{(L)}(y, y) \end{array} \right] \right) \,. \tag{1.6}$$

For a given neural network, defined as before, its Neural Tangent Kernel (NTK) is the matrix valued Kernel whose components $\Theta_{\mu\nu}^{(L)} : \mathbb{R}^{n_0} \times \mathbb{R}^{n_0} \to \mathbb{R}$ are defined by

$$\Theta_{\mu\nu}^{(L)}(x, y) = \sum_{\theta \in \mathcal{P}} \frac{\partial f_{\theta,\mu}^{(L)}}{\partial \theta}(x) \frac{\partial f_{\theta,\nu}^{(L)}}{\partial \theta}(y) \,, \tag{1.7}$$

with $\mathcal{P} = \{W_{ij}^{(\ell)}, b_k^{(\ell)}\}$ the set of all (learnable) parameters.

A fundamental observation by Jacot et al. (2018), significantly deepened in (Arora et al., 2018), (Yang, 2019), is that, under the previous conditions, as $n_1, \ldots, n_{L-1} \to \infty$, the NTK converges in law to a deterministic kernel

$$\Theta_{\mu\nu}^{(L)} \to \Theta_{\infty,\mu\nu}^{(L)} = \Theta_\infty^{(L)} \delta_{\mu\nu} \,, \tag{1.8}$$

with the scalar kernel $\Theta_\infty^{(L)} : \mathbb{R}^{n_0} \times \mathbb{R}^{n_0} \to \mathbb{R}$ defined recursively by

$$\Theta_\infty^{(1)}(x, y) = \frac{1}{n_0} x^\intercal y + \beta^2 \,, \tag{1.9}$$

$$\Theta_\infty^{(\ell+1)}(x, y) = \Theta_\infty^{(\ell)}(x, y) \dot{\Sigma}^{(\ell+1)}(x, y) + \Sigma^{(\ell+1)}(x, y) \,, \tag{1.10}$$

where, for $\ell \geq 1$,

$$\Sigma^{(\ell+1)}(x, y) = \mathbb{E}_{f \sim \hat{\Sigma}^{(\ell)}} \left[ \sigma(f(x)) \, \sigma(f(y)) \right] + \beta^2 \,, \tag{1.11}$$

$$\dot{\Sigma}^{(\ell+1)}(x, y) = \rho_W^2 \mathbb{E}_{f \sim \hat{\Sigma}^{(\ell)}} \left[ \dot{\sigma}(f(x)) \, \dot{\sigma}(f(y)) \right] \,. \tag{1.12}$$

## 1.2 MAIN RESULTS

Recall that a symmetric matrix $P \in \mathbb{R}^{N \times N}$ is strictly positive definite provided that $u^\intercal P u > 0$ , for all $u \in \mathbb{R}^N \setminus \{0\}$. Recall also the following:

**Definition 1.** *A symmetric function*

$$K : \mathbb{R}^{n_0} \times \mathbb{R}^{n_0} \to \mathbb{R}$$

*is a strictly positive definite Kernel provided that, for all choices of finite subsets of $\mathbb{R}^{n_0}$, $X = \{x_1, \ldots, x_N\}$ (thus without repeated elements), the matrix*

$$K_X := \left[ K(x_i, x_j) \right]_{i,j \in \{1, \ldots, N\}} \,, \tag{1.13}$$

*is strictly positive definite.*

We are now ready to state our main results.

**Theorem 1** (Positivity of the NTK for networks with biases)**.** *Consider an architecture with activated biases, i.e. $\beta \neq 0$, and a continuous, almost everywhere differentiable and non-polynomial activation function $\sigma$. Then, the NTK $\Theta_\infty^{(L)}$ is (in the sense of Definition 1) a strictly positive definite Kernel for all $L \geq 2$.*

**Remark 1.** *Notice that the previous result is sharp in the following sense. First, the NTK matrix clearly degenerates if we have repeated training inputs, so, in practice, our result does not make any spurious restrictions at the level of the data. Second, it is also known, see for instance Panigrahi et al. (2020, theorem 4.3) and compare with Jacot et al. (2018, Remark 5), that the minimum eigenvalue of an NTK matrix is zero if the activation function is polynomial and the data set is sufficiently large. Finally, the regularity assumption of almost everywhere differentiability is, in view of (1.12), required to have a well defined NTK.*

For the sake of completeness we will also establish a positivity result for the case with no biases ($\beta = 0$). This situation calls for extra work and stronger, yet still reasonable, assumptions about the training set. This further emphasizes the well-established relevance of including biases in our models.

**Theorem 2** (Positivity of the NTK for networks with no biases)**.** *Consider an architecture with deactivated biases ($\beta = 0$) and a continuous, almost everywhere differentiable and non-polynomial activation function $\sigma$. If the training inputs $\{x_1, \ldots, x_N\}$ are all pairwise non-proportional, then, for all $L \geq 2$, the matrix $\Theta_X^{(L)} = \left[ \Theta_\infty^{(L)}(x_i, x_j) \right]_{i,j \in [N]}$ is strictly positive definite.*

**Remark 2.** *As it is well known, one of the main effects of adding bias terms corresponds, in essence, to adding a new dimension to the input space and embedding the inputs into the hyperplane $x_{n_0+1} = 1$. This has the effect of turning distinct inputs, in the original $\mathbb{R}^{n_0}$ space, into non-proportional inputs in $\mathbb{R}^{n_0+1}$. Hopefully this sheds some light into the distinctions between the last two theorems.*

The proofs of the previous theorems are the subject of subsection 2.2 (see Corollaries 2 and 4 respectively). They rely partially on the following interesting characterization of polynomial functions which we take the opportunity to highlight here:

**Theorem 3.** *Let $z = (z_i)_{i \in [N]}$, $w = (w_i)_{i \in [N]} \in \mathbb{R}^N$ be totally non-aligned, meaning that*

$$\begin{vmatrix} z_i & w_i \\ z_j & w_j \end{vmatrix} \neq 0 \ , \text{ for all } i \neq j \ , \tag{1.14}$$

*and let $\sigma : \mathbb{R} \to \mathbb{R}$ . If there exists $u \in \mathbb{R}^N \setminus \{0\}$, such that*

$$\sum_{i=1}^N u_i \, \sigma(\theta_1 z_i + \theta_2 w_i) = 0 \ , \text{ for every } (\theta_1, \theta_2) \in \mathbb{R}^2 \ , \tag{1.15}$$

*then $\sigma$ is a polynomial.*

The previous result is an immediate consequence of Theorem 7 and Theorem 6, proven in the appendix.

### 1.3 RELATED WORK

In their original work, Jacot et al. (2018) already discussed the issue studied in the current paper and proved that, under the additional hypothesis that the training data lies in the unit sphere, the Neural Tangent Kernel (NTK) is strictly positive definite for Lipschitz activation functions. Du et al. (2019) made a further interesting contribution in the case where there are no biases. They found that if the activation function is analytic but non-polynomial and no two data points are parallel, then the minimum eigenvalue of an appropriate Gram matrix is positive; this, in particular, provides a positivity result for the NTK, under the described restrictions. As mentioned above we generalize this result by withdrawing these and other restrictions.

Later Allen-Zhu et al. (2019) worked with the specific case of ReLu activation functions, but were able to drop the very restrictive hypothesis that the data points all lie in the unit sphere. Instead, they provide a result showing that for ReLu activation functions, the minimum eigenvalue of the NTK is

"large" under the assumption that the data is $\delta$-separated (meaning that no two data points are very close). In a related work, Panigrahi et al. (2020) conducted a study on one hidden layer neural nets where only the input layer is trained. They made the assumption that the data points are on the unit sphere and satisfy a specific $\delta$-separation condition. Their results are applicable to large networks where the number of neurons $m$ increases with the number of data points. Moreover, if the activation function is polynomial the minimal eigenvalue of the NTK vanishes for large enough data sets, as illustrated in theorem 4.3 of the same reference. This shows that our conditions, at the level of the activation function, are also necessary so that our results are sharp.

There are a number of other works which investigate these problems and come to interesting partial results. They all have some intersection with the above mentioned results, but given their relevance we shall briefly mention some of these below.

In Lai et al. (2023) it is shown that the NTK is strictly positive definite for a two-layered neural net with ReLU activation. Later, Li et al. (2023) extended this result to multilayered nets, but maintained the ReLU activation restriction. Montanari & Zhong (2023) studied the NTK of neural nets with a special linear architecture. They proved that, in this specific framework, the NTK is strictly positive definite . Bombari et al. (2022) found a lower bound on the smallest eigenvalue of the empirical NTK for finite deep neural networks, where at least one hidden layer is large, with the number of neurons growing linearly with amount of data. They also require a Lipschitz activation function with Lipschitz derivative. Related results can also be found in (Banerjee et al., 2023) and references therein. Other interesting and relevant works which study the positivity of the NTK and/or its eigenvalues include (Zhu et al., 2022; Fan & Wang, 2020; Nguyen et al., 2020).

## 2 THE POSITIVITY OF THE NTK

### 2.1 WARM-UP: AN INSTRUCTIVE SPECIAL CASE

It might be instructive to first consider the simplest of cases: a one hidden layer network $L = 2$, with one-dimensional inputs $n_0 = 1$ and one-dimensional outputs $n_2 = 1$. This will allow us to clarify part of the strategy employed in the proof of the general case that will be presented in the next section; nonetheless, the more impatient reader can skip the present section.

In this special case we do not even need to use the recurrence relation (1.9)-(1.10), since we can compute the NTK directly from its definition (1.7). In order to do that it is convenient to introduce the *perceptron random variable*

$$p(x) = W^{(1)}\sigma(W^{(0)}x + b^{(0)}) , \tag{2.1}$$

with parameters $\theta \in \{W^{(0)}, b^{(0)}, W^{(1)}\}$ satisfying (1.3), and the *kernel random variable*

$$\mathcal{K}_\theta(x, y) := \sum_{\theta \in \{W^{(0)}, b^{(0)}, W^{(1)}\}} \frac{\partial p}{\partial \theta}(x) \frac{\partial p}{\partial \theta}(y) . \tag{2.2}$$

Using perceptrons the networks under analysis in this section are functions $f_\theta^{(2)} : \mathbb{R} \to \mathbb{R}$ that can be written as

$$f_\theta^{(2)}(x) = \frac{1}{\sqrt{n_1}} \sum_{k=1}^{n_1} p_k(x) + \beta\, b^{(1)} , \tag{2.3}$$

where $n_1$ is the number of neurons in the hidden layer and where each perceptron has *iid* parameters $\theta_k \in \mathcal{P}_k = \{W_k^{(0)}, b_k^{(0)}, W_k^{(1)}\}$ and $b^{(1)}$ that satisfy (1.3). Moreover the corresponding NTK (1.7), which in this case is a scalar, satisfies

$$\Theta^{(2)}(x, y) = \frac{1}{n_1} \sum_{k=1}^{n_1} \sum_{\theta_k \in \mathcal{P}_k} \frac{\partial p_k(x)}{\partial \theta_k} \frac{\partial p_k(y)}{\partial \theta_k} + \frac{\partial f_\theta^{(2)}(x)}{\partial b^{(1)}} \frac{\partial f_\theta^{(2)}(y)}{\partial b^{(1)}}$$

$$= \frac{1}{n_1} \sum_{k=1}^{n_1} \mathcal{K}_{\theta_k}(x, y) + \beta^2 .$$

In the limit $n_1 \to \infty$, the Law of Large Numbers guarantees that it converges a.s. to

$$\Theta_\infty^{(2)}(x, y) = \mathbb{E}_\theta\left[\mathcal{K}_\theta(x, y)\right] + \beta^2 . \tag{2.4}$$

If we denote the gradient of $p$, with respect to $\theta$, at $x \in \mathbb{R}$, by

$$\nabla_\theta p(x)^\top = \left[ \frac{\partial p}{\partial W^{(0)}}(x) , \ \frac{\partial p}{\partial b^{(0)}}(x) , \ \frac{\partial p}{\partial W^{(1)}}(x) \right]$$

$$= \left[ x W^{(1)} \dot{\sigma}(W^{(0)} x + b^{(0)}) , \ W^{(1)} \dot{\sigma}(W^{(0)} x + b^{(0)}) , \ \sigma(W^{(0)} x + b^{(0)}) \right] , \qquad (2.5)$$

we see that $\mathcal{K}_\theta(x, y) = \nabla_\theta p(y)^\top \nabla_\theta p(x)$. Then the Gram matrix $(\mathcal{K}_\theta)_X$, defined over the training set $X = \{x_1, \ldots, x_N\}$, is

$$(\mathcal{K}_\theta)_X := \left[ \mathcal{K}_\theta(x_i, x_j) \right]_{i,j \in [n]} = \nabla_\theta p(X)^\top \nabla_\theta p(X) ,$$

where we used the $3 \times N$ matrix $\nabla_\theta p(X)$ given by

$$\nabla_\theta p(X) = \begin{pmatrix} \nabla_\theta p(x_1) & \nabla_\theta p(x_2) & \cdots & \nabla_\theta p(x_N) \end{pmatrix} .$$

The (infinite width) NTK matrix over $X$ is defined by $\Theta_X^{(2)} := \left[ \Theta_\infty^{(2)}(x_i, x_j) \right]_{i,j \in [n]}$ and, in view of (2.4), the two matrices are related by

$$\Theta_X^{(2)} = \mathbb{E}_\theta \left[ (\mathcal{K}_\theta)_X \right] + \beta^2 e e^\top , \qquad (2.6)$$

where $e := [1 \cdots 1]^\top \in \mathbb{R}^N$. Now, given $u \in \mathbb{R}^N$ we have

$$u^\top \Theta_X^{(2)} u = \mathbb{E}_\theta \left[ u^\top \nabla_\theta p(X)^\top \nabla_\theta p(X) u \right] + \beta^2 u^\top e e^\top u = \mathbb{E}_\theta \left[ (\nabla_\theta p(X) u)^\top \nabla_\theta p(X) u \right] + \beta^2 (u^\top e)^2$$

$$= \mathbb{E}_\theta \left[ \left\| \nabla_\theta p(X) u \right\|^2 \right] + \beta^2 (u^\top e)^2 \geq 0 , \qquad (2.7)$$

which shows that $\Theta_X^{(2)}$ is positive semi-definite.

Moreover, we can use the previous observations to achieve our main goal for this section by showing that, under slightly stronger assumptions, $\Theta_X^{(2)}$ is, in fact, strictly positive definite. To do that we will only need to assume that there are no repeated elements in the training set and that the activation function $\sigma$ is continuous, non-polynomial and almost everywhere differentiable with respect to the Lebesgue measure. Under such conditions, assume there exists $u \neq 0$ such that $u^\top \Theta_X^{(2)} u = 0$, i.e., that the NTK matrix is not strictly positive definite. From (2.7) we see that this can only happen if $\beta u^\top e = 0$ and $\nabla_\theta p(X) u = 0$, for almost every $\theta$, as measured by the parameter initialization. However, as $\nabla_\theta p(\cdot)$ is continuous and our parameters are sampled from probability measures with full support, we conclude that

$$\nabla_\theta p(X) u = 0 \ , \ \text{for all } \theta \in \mathbb{R}^3 .$$

In particular, the third component of this vector also vanishes, a condition which, in view of (2.5), can be cast in the form

$$\sum_{i=1}^N u_i \sigma(W^{(0)} x_i + b^{(0)}) = 0 \ , \ \text{for all } (W^{(0)}, b^{(0)}) \in \mathbb{R}^2 .$$

It follows from Theorem 3 that this condition implies that $\sigma$ must be a polynomial, in contradiction with our assumptions. The proof of this theorem assuming only the continuity of $\sigma$ is rather involved and will be postponed to Appendix A. For now, in coherence with the pedagogical spirit of this section, we will content ourselves with a simple proof that holds for the case of an activation function which is $C^{N-1}$. Note, however, that this is insufficient for many application in deep learning, where one uses activation functions which fail to be differentiable at some points; the ReLu being the prime example. Let $u \neq 0$ satisfy (1.15). If $u$ has any vanishing components these can be discarded from (1.15), so we can assume, without loss of generality, that all $u_i \neq 0$. Therefore we are allowed to rewrite (1.15) in the following form

$$\sigma(\theta_1 z_N + \theta_2 w_N) = \sum_{i=1}^{N-1} u_i^{(1)} \sigma(\theta_1 z_i + \theta_2 w_i) ,$$

let $u \neq 0$ satisfy (1.15). If $u$ has any vanishing components these can be discarded from (1.15), so we can assume, without loss of generality, that all $u_i \neq 0$. Therefore we are allowed to rewrite (1.15) in the following form

$$\sigma(\theta_1 z_N + \theta_2 w_N) = \sum_{i=1}^{N-1} u_i^{(1)} \sigma(\theta_1 z_i + \theta_2 w_i) ,$$

where all $u_i^{(1)} := u_i/u_N \neq 0$. Now we differentiate (2.8) with respect to $\theta_1$ and with respect to $\theta_2$ in order to obtain

$$z_N\dot{\sigma}(\theta_1 z_N + \theta_2 w_N) = \sum_{i=1}^{N-1} u_i^{(1)} z_i \dot{\sigma}(\theta_1 z_i + \theta_2 w_i) , \tag{2.8}$$

$$w_N\dot{\sigma}(\theta_1 z_N + \theta_2 w_N) = \sum_{i=1}^{N-1} u_i^{(1)} w_i \dot{\sigma}(\theta_1 z_i + \theta_2 w_i). \tag{2.9}$$

Then, we multiply equations 2.8 and 2.9 by $w_N$ and $z_N$ respectively and subtract them to derive

$$\sum_{i=1}^{N-1} (z_N w_i - w_N z_i) u_i^{(1)} \dot{\sigma}(\theta_1 z_i + \theta_2 w_i) = 0 .$$

Since $(z_N w_{N-1} - w_N z_{N-1}) u_{N-1}^{(1)} \neq 0$, we have

$$\dot{\sigma}(\theta_1 z_{N-1} + \theta_2 w_{N-1}) = \sum_{i=1}^{N-2} u_i^{(2)} \dot{\sigma}(\theta_1 z_i + \theta_2 w_i) ,$$

where $u_i^{(2)} := -\frac{(z_N w_i - w_N z_i) u_i^{(1)}}{(z_N w_{N-1} - w_N z_{N-1}) u_{N-1}^{(1)}} \neq 0$. Once again we differentiate with respect to $\theta_1$ and $\theta_2$ and equate the results to obtain

$$\sum_{i=1}^{N-2} (z_{N-1} w_i - w_{N-1} z_i) u_i^{(2)} \ddot{\sigma}(\theta_1 z_i + \theta_2 w_i) = 0 .$$

Under the assumed conditions we can keep on repeating this process until we arrive at

$$(z_2 w_1 - w_2 z_1) u_i^{(N-1)} \sigma^{(N-1)}(\theta_1 z_1 + \theta_2 w_1) = 0 .$$

Since the last equality holds for all $(\theta_1, \theta_2) \in \mathbb{R}^2$ we conclude that $\sigma^{(N-1)} \equiv 0$ which implies that $\sigma$ is a polynomial.

## 2.2 The general case

In this section we will consider general networks (1.1), in terms of the number of inputs, outputs and hidden layers, and we will show that, under very general assumptions, the (infinite width limit) NTK, $\Theta_\infty^{(L)}$, is strictly positive definite, for all $L \geq 2$ (at least one hidden layer). This will be achieved by studying the positive definiteness of various symmetric matrices related to the recurrence formulas (1.9)-(1.10).

Given a symmetric function $K : \mathbb{R}^{n_0} \times \mathbb{R}^{n_0} \to \mathbb{R}$, and a training set $X = \{x_1, \ldots, x_N\} \subset \mathbb{R}^{n_0}$, we define its matrix over $X$ by

$$K_X = \left[ K_{ij} := K(x_i, x_j) \right]_{i,j \in [N]} , \tag{2.10}$$

where we use the notation $[N] = \{1, \ldots, N\}$. Furthermore, to clarify our terminology, recall that the symmetric matrix $K_X$ is strictly positive definite when $u^\intercal K_X u > 0$, for all $u \in \mathbb{R}^N \setminus \{0\}$.

Inspired by the recurrence structure in both (1.4), 1.5 and (1.9), (1.10), we consider two Kernel matrices over $X$ related by the identity

$$K_{ij}^{(2)} = \mathbb{E}_{f \sim K^{(1)}} \left[ \sigma\big(f(x_i)\big) \sigma\big(f(x_j)\big) \right] + \beta^2 . \tag{2.11}$$

In the following, given $f : \mathbb{R}^{n_0} \to \mathbb{R}$, we will write $Y = f(X) = [f(x_1) \cdots f(x_N)]^\intercal \in \mathbb{R}^N$ and, as before, we will also use the notation $e := [1 \cdots 1]^\intercal \in \mathbb{R}^N$. Recall that the notation $\sim K^{(1)}$, introduced after (1.6), is a shorthand for a centered Gaussian Process with covariance function $K^{(1)}$. Analogously, $\sim K_X^{(1)}$ refers to the centered normal distribution with covariance matrix $K_X^{(1)}$. So, when $f \sim K^{(1)}$ then $f(X) \sim K_X^{(1)}$. We are assuming $K_X^{(1)}$ is positive semi-definite.

We see that, for $i, j \in [N]$, the $(i, j)$ entry of $K_X^{(2)}$ is given by (2.11), and so we can write it as

$$K_X^{(2)} = \mathbb{E}_{f(X) \sim K_X^{(1)}}\left(\sigma\big(f(X)\big)\sigma\big(f(X)^\top\big)\right) + \beta^2 e\, e^\top = \mathbb{E}_{Y \sim K_X^{(1)}}\left(\sigma(Y)\sigma(Y)^\top\right) + \beta^2 e\, e^\top$$

$$= \mathbb{E}_{Y \sim K_X^{(1)}}\left(\begin{bmatrix}\sigma(Y) & \beta e\end{bmatrix}\begin{bmatrix}\sigma(Y)^\top \\ \beta e^\top\end{bmatrix}\right),$$

where $\sigma\big(f(X)\big)$ and $\sigma(Y)$ are $N \times 1$ matrices defined by

$$\sigma\big(f(X)\big) = \begin{bmatrix}\sigma(f(x_1)) \cdots \sigma(f(x_N))\end{bmatrix}^\top \quad \text{and} \quad \sigma(Y) = \begin{bmatrix}\sigma(y_1) \cdots \sigma(y_N)\end{bmatrix}^\top .$$

Now, given $u \in \mathbb{R}^N$, we have

$$u^\top K_X^{(2)} u = \mathbb{E}_{Y \sim K_X^{(1)}}\left(u^\top \begin{bmatrix}\sigma(Y) & \beta e\end{bmatrix}\begin{bmatrix}\sigma(Y)^\top \\ \beta e^\top\end{bmatrix} u\right) = \mathbb{E}_{Y \sim K_X^{(1)}}\left(\begin{bmatrix}u^\top\sigma(Y) & \beta u^\top e\end{bmatrix}\begin{bmatrix}\sigma(Y)^\top u \\ \beta e^\top u\end{bmatrix}\right)$$

$$= \mathbb{E}_{Y \sim K_X^{(1)}}\left(\left\|[\sigma(Y)^\top u\,,\ \beta e^\top u]\right\|^2\right) . \tag{2.12}$$

We conclude that $u^\top K_X^{(2)} u \geq 0$, that is, $K_X^{(1)}$ positive semi-definite implies that $K_X^{(2)}$ is also positive semi-definite. As already observed in Jacot et al. (2018), it turns out that we can easily strengthen this relation:

**Proposition 1** (Induction step)**.** *Assume that the activation function $\sigma$ is continuous and not a constant. If $K_X^{(1)}$ is strictly positive definite, then $K_X^{(2)}$, defined by* (2.11)*, is also strictly positive definite.*

*Proof.* Assume that under the prescribed assumptions $K_X^{(2)}$ is not strictly positive definite. Then there exists $u \in \mathbb{R}^N \setminus \{0\}$ such that $u^\top K_X^{(2)} u = 0$. In view of (2.12) this implies that $\sigma(Y)^\top u = 0$, $\mathcal{N}(0, K_X^{(1)})$-almost everywhere, but since $K_X^{(1)}$ is, by assumption, strictly positive definite, the corresponding Gaussian measure has full support in $\mathbb{R}^N$ and, by continuity, we must have

$$\sigma(y)^\top u = 0 \quad, \text{ for all } y \in \mathbb{R}^N . \tag{2.13}$$

By rearranging the components of $u$ we can assume that $u_N \neq 0$. Then, the last identity, applied to a vector of the form $y = (0, \dots, 0, x)$, $x \in \mathbb{R}$, would imply that $\sigma(x) = -\sigma(0) \sum_{i=1}^{N-1} \frac{u_i}{u_N}$, which shows that, under the previous circumstances, $\sigma$ must be a constant. Since this contradicts our assumptions we conclude that $K_X^{(2)}$ is strictly positive definite. $\qquad\square$

While Proposition 1 offers the necessary induction step to propagate the favorable sign to the matrices $\hat{\Sigma}_X^{(\ell+1)}$ and $\Sigma_X^{(\ell+1)}$, it is insufficient to assert that these matrices are strictly positive definite. Since $\hat{\Sigma}_X^{(1)}$ typically does not exhibit this property. Nonetheless, we will now show that under suitable and relatively mild conditions related to the training set and activation function, the desired positivity for $\hat{\Sigma}_X^{(2)}$ and $\Sigma_X^{(2)}$ emerges from the recurrence relations (1.5) and (1.11).

### 2.2.1 NETWORKS WITH BIASES

We will first deal with the case with biases ($\beta \neq 0$). Our strategy, inspired by the special case studied in the previous section, will be to steer towards Theorem 3 to obtain the desired conclusion.

**Theorem 4.** *Assume that the training inputs $x_i$ are all distinct, and that the activation function $\sigma$ is continuous and non-polynomial. If*

$$K^{(1)}(x, y) = \alpha^2 x^\top y + \beta^2 , \tag{2.14}$$

*with $\alpha\beta \neq 0$, then $K_X^{(2)}$, as defined by* (2.11) *is strictly positive definite.*

*Proof.* As in the proof of Proposition 1, if $K_X^{(2)}$ is not strictly positive definite, then there exists a non-vanishing $u \in \mathbb{R}^N$ such that $\sigma(Y)^\top u = 0$, $\mathcal{N}(0, K_X^{(1)})$-almost everywhere. Let $X$ also denote the matrix whose columns are the training inputs. Then, for $\tilde{X} = \begin{bmatrix}\alpha X^\top & \beta e\end{bmatrix}$ we can write

$$K_X^{(1)} = \alpha^2 X^\top X + \beta^2 e\, e^\top = \begin{bmatrix}\alpha X^\top & \beta e\end{bmatrix}\begin{bmatrix}\alpha X \\ \beta e^\top\end{bmatrix} = \tilde{X}\tilde{X}^\top .$$

Let $\mathrm{rank}(\tilde{X}) = r \geq 1$ and $\tilde{X}_{(r)}$ be a $r \times (n_0 + 1)$ matrix containing $r$ linearly independent rows of $\tilde{X}$. We assume without loss of generality that $\tilde{X}_{(r)}$ consists of the first $r$ rows of $\tilde{X}$. Then, there exists an $N \times r$ matrix $B$ such that

$$\tilde{X} = B\tilde{X}_{(r)} \ . \tag{2.15}$$

The distribution, over $\mathbb{R}^N$, of $Y = (Y_1 \ldots, Y_r, \ldots, Y_N) \sim \mathcal{N}(0, K_X^{(1)})$, in general, has a degenerated covariance, however the distribution over the first $r$ components $Y_{(r)} := (Y_1, \ldots, Y_r) \sim \mathcal{N}(0, \tilde{X}_{(r)}\tilde{X}_{(r)}^\top)$ has a non degenerated covariance matrix and

$$\mathrm{Cov}(BY_{(r)}) = B \, \mathrm{Cov}(Y_{(r)})B^\top = B\tilde{X}_{(r)}\tilde{X}_{(r)}^\top B^\top = \tilde{X}\tilde{X}^\top = K_X^{(1)}.$$

Thus the fact that there exists $u \in \mathbb{R}^N \setminus \{0\}$ such that $\sigma(Y)^\top u = 0$, for $\mathcal{N}(0, K_X^{(1)})$-almost every $Y \in \mathbb{R}^N$, is equivalent to $\sigma(BY_{(r)})^\top u = 0$, for $\mathcal{N}(0, \tilde{X}_{(r)}\tilde{X}_{(r)}^\top)$-almost every $Y_{(r)} \in \mathbb{R}^r$. An advantage of the last formulation is that the corresponding measure has full support in $\mathbb{R}^r$ so, by continuity, we conclude that

$$\sigma(By)^\top u = 0 \ , \ \text{for all } y \in \mathbb{R}^r \ . \tag{2.16}$$

To proceed we will need the following

**Lemma 1.** *Assume that $B$ is an $N \times r$ matrix with no repeated rows. Then there exists $y^{\neq} \in \mathbb{R}^r$ such that $z^{\neq} = By^{\neq}$ is a vector in $\mathbb{R}^N$ with pairwise distinct entries.*

*Proof.* Let $y^{\neq}(x) = (1, x, \ldots, x^{r-1})$, for $x \in \mathbb{R}$. Then $z^{\neq}(x) := By^{\neq}(x)$ is a vector whose entries are polynomials $p_i(x) = \sum_j B_{ij}x^{j-1}$, $i \in [N]$, which (as polynomials) are pairwise distinct since the rows of $B$ are also pairwise distinct.

Now consider the set $\mathcal{I}$, of real numbers where at least two of the polynomials coincide, and the sets $\mathcal{I}_{ij}$, corresponding to the solutions of $p_i(x) = p_j(x)$, for a specific pair of indices $i \neq j$. Clearly $\mathcal{I} \subset \cup_{i \neq j}\mathcal{I}_{ij}$, and therefore

$$\#\mathcal{I} \leq \sum_{i \neq j} \#\mathcal{I}_{ij} \leq \binom{N}{2} \times (r-1) < \#\mathbb{R} \ .$$

In conclusion, we can choose $x \in \mathbb{R}$ such that all the entries of $z^{\neq}(x) = (p_1(x), \ldots, p_N(x))$ are pairwise distinct, and we are done. $\qquad\square$

Now, since $X$ has no repeated elements, $B$ has no repeated rows. Then, for any given $(\theta_1, \theta_2)$, let $y = \theta_1 y^{\neq} + \theta_2 \beta e$, with $y^{\neq}$ as in the previous lemma, for which $By = \theta_1 z^{\neq} + \theta_2 \beta e$. In such case the equality (2.16) becomes $\sum_{i=1}^N u_i\sigma(\theta_1 z_i + \theta_2 \beta) = 0$, for all $(\theta_1, \theta_2) \in \mathbb{R}^2$, with the $z_i$ pairwise distinct and $\beta \neq 0$. But since a vector $z^{\neq} \in \mathbb{R}^N$ with all entries pairwise distinct is, in the sense of (1.14), *totally non-aligned* with the bias vector $e = (1, \ldots, 1)$, we can apply Theorem 3 to conclude that $\sigma$ must be a polynomial. This contradicts our assumptions, therefore $K_X^{(2)}$ must be strictly positive definite. $\qquad\square$

An immediate consequence of the preceding result is that $\hat{\Sigma}_X^{(2)}$ and $\Sigma_X^{(2)}$ are strictly positive definite. The induction provided by Proposition 1 then leads to the following conclusion.

**Corollary 1.** *Under the conditions of Theorem 4, and for all $\ell \geq 2$, $\hat{\Sigma}_X^{(\ell)}$ and $\Sigma_X^{(\ell)}$, defined by (1.4), (1.5) and (1.11), are strictly positive definite.*

We are now ready to achieve our main goal for this section, which is the proof of Theorem 1 which for convenience we restate here as follows.

**Corollary 2** (Theorem 1). *Under the conditions of Theorem 4, and for $\sigma$ differentiable almost everywhere, the matrix $\Theta_X^{(\ell)}$ is strictly positive definite, for all $\ell \geq 2$.*

*Proof.* We start by noticing that, if we assume that $\sigma$ is differentiable almost everywhere, a computation similar to the one used for equation (2.12), shows that $\dot{\Sigma}_X^{(\ell)} = [\dot{\Sigma}^{(\ell)}(x_i, x_j)]_{i,j \in \{1,\ldots,N\}}$ is positive semi-definite, for all $\ell \geq 2$. On the other hand $\Theta_X^{(1)}$ as defined in remark 1 is positive semi-definite. We recall that due to the Schur product theorem, the Hadamard product of

two positive semi-definite matrices remains positive semi-definite. Thus, the matrix $\Theta_X^{(1)} \odot \dot{\Sigma}_X^{(2)} = \left[ \Theta_\infty^{(1)}(x_i, x_j) \dot{\Sigma}_\infty^{(2)}(x_i, x_j) \right]_{i,j \in [N]}$ is positive semi-definite. Since the sum of a strictly positive definite matrix with a positive semi-definite matrix gives rise to a strictly positive definite matrix, we can conclude that $\Theta_X^{(2)} = \Theta_X^{(1)} \odot \dot{\Sigma}_X^{(2)} + \Sigma_X^{(2)}$ is strictly positive definite. The statement then follows from the recurrence (1.10) and Corollary 1. □

### 2.2.2 Networks with no biases

We can also deal with the case with no biases, i.e., $\beta = 0$, but this case requires more effort and stronger (although still mild) assumptions on the training set; another testimony in favor of the well known importance of including biases in our models.

**Theorem 5.** *Assume that the training inputs are all pairwise non-proportional and that the activation function $\sigma$ is continuous and non-polynomial. If $K^{(1)}(x, y) = \alpha^2 x^\intercal y$, with $\alpha \neq 0$, then $K_X^{(2)}$, as defined by (2.11) is strictly positive definite.*

*Proof.* Just as in the proof of Theorem 4 we can construct a ranked $r$ matrix $\tilde{X}_{(r)}$ and a matrix $B$ such that (2.15) holds, but now, with the removal of the biases column from $\tilde{X} \in \mathbb{R}^{N \times n_0}$. Although we do not have the helpful bias column, our assumptions on the training set guarantee that the rows of $B$ are pairwise non-proportional which allows to prove the following:

**Lemma 2.** *Assume the rows of $B$ are all pairwise non-proportional, then there exists $y_1, y_2 \in \mathbb{R}^r$, such that the $\mathbb{R}^N$ vectors $z = By_1$ and $w = By_2$ are* totally non-aligned, *meaning that (1.14) holds.*

*Proof.* As before, consider the polynomials defined by $p_i(x) = \sum_j B_{ij} x^{j-1}$, which (as polynomials) are pairwise non-proportional, in view of the assumptions on $B$. Now choose $x_1$ such that $w = (w_i := p_i(x_1))_{i \in [N]}$ has all non zero entries and consider the polynomials $q_i = p_i / w_i$ which (as polynomials) are distinct, since the $p_i$ are pairwise non-proportional. We then see that

$$\begin{vmatrix} p_i(x_2) & w_i \\ p_j(x_2) & w_j \end{vmatrix} = w_j p_i(x_2) - w_i p_j(x_2) \neq 0 \Leftrightarrow q_i(x_2) \neq q_j(x_2) . \tag{2.17}$$

So we can construct the desired $z$ by setting $z_i = p_i(x_2)$, where $x_2$ is such that all $q_i(x_2)$ are distinct. □

Given $(\theta_1, \theta_2)$, let $y = \theta_1 y_1 + \theta_2 y_2$, with the $y_i$ as in the previous lemma. Then, $By = \theta_1 z + \theta_2 w$ and the equality (2.16) becomes $\sum_{i=1}^N u_i \sigma(\theta_1 z_i + \theta_2 w_i) = 0$, for all $(\theta_1, \theta_2) \in \mathbb{R}^2$, with $z = (z_i)$ and $w = (w_i)$ totally non-aligned. In view of Theorem 3 $\sigma$ must be a polynomial. Once more, this contradicts our assumptions, therefore $K_X^{(2)}$ must be strictly positive definite. □

An immediate consequence of Proposition 1 and the previous result is the following

**Corollary 3.** *Under the conditions of Theorem 5, for all $\ell \geq 2$, $\hat{\Sigma}_X^{(\ell)}$ and $\Sigma_X^{(\ell)}$, defined by (1.4), (1.5), and (1.11), with $\beta = 0$, are strictly positive definite.*

Finally, as in the case in which $\beta \neq 0$, we are now ready to conclude that:

**Corollary 4** (Theorem 2). *Under the conditions of Theorem 5, assume moreover that $\sigma$ is differentiable almost everywhere and $\beta = 0$. Then, $\Theta_X^{(\ell)}$ is strictly positive definite, for all $\ell \geq 2$.*

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

# A  TWO CHARACTERIZATIONS OF POLYNOMIAL FUNCTIONS.

In Section 2.1 we proved Theorem 3, in the simple case when $\sigma$ is $C^{N-1}$, by showing that, in such case, the conditions of the theorem implied that $\sigma^{(N-1)} \equiv 0$, from which one immediately concludes that $\sigma$ must be a polynomial. In this section, we will show how to extend this result to the case when $\sigma$ is only continuous. For that we clearly need different techniques. Basically we will rely in the analysis of $\sigma$'s finite differences and show that, under the conditions of the theorem, all finite differences of order $N - 1$ vanish. Remarkably this also implies that $\sigma$ is a polynomial.

More precisely, in theorem 7 we will show that, under the conditions of Theorem 3, we must have $\Delta_h^{N-1}\sigma(x) = 0$, for all $x$ and $h$, and in Theorem 6 that if a continuous function $\sigma$ satisfies this relation then $\sigma$ must be a polynomial. We believe that this last result is most likely already well known, unfortunately we were unable to find it in the literature, so we present a complete proof here.

For a given function $f : \mathbb{R} \to \mathbb{R}$ let its finite differences be given by

$$(\Delta_h f)(x) = f(x + h) - f(x) . \tag{A.1}$$

Note that each finite difference $\Delta_h$ is a linear operator on the space of functions $\mathcal{M} := \mathrm{Map}(\mathbb{R}, \mathbb{R})$, $\Delta_h : \mathcal{M} \to \mathcal{M}$.

The finite differences of second order with increments $\boldsymbol{h} = (h_1, h_2)$ are defined by

$$\begin{aligned}
(\Delta_{\boldsymbol{h}}^2 f)(x) &= \left(\Delta_{h_2}(\Delta_{h_1} f)\right)(x) \\
&= (\Delta_{h_1} f)(x + h_2) - (\Delta_{h_1} f)(x) \\
&= \left(f(x + h_2 + h_1) - f(x + h_2)\right) - (f(x + h_1) - f(x)) .
\end{aligned}$$

Note that $\Delta_{h_1}$ and $\Delta_{h_2}$ commute, that is $\Delta_{h_1}(\Delta_{h_2} f) = \Delta_{h_2}(\Delta_{h_1} f)$. Proceeding inductively we have that

$$\Delta_{(\boldsymbol{h}, h_{n+1})}^{n+1} f(x) = \Delta_{h_{n+1}}(\Delta_{\boldsymbol{h}}^n f)(x) = (\Delta_{\boldsymbol{h}}^n f)(x + h_{n+1}) - (\Delta_{\boldsymbol{h}}^n f)(x).$$

When $\boldsymbol{h} = (h, \dots, h)$ we have $\Delta_{\boldsymbol{h}}^n = \Delta_h^n$.

**Theorem 6.** *Let $f : \mathbb{R} \to \mathbb{R}$ be a function that, for a given $n \in \mathbb{N}$, satisfies $\Delta_h^n f(x) = 0$, for all $h$ and $x$. Then $f$ is a polynomial of order $n - 1$.*

*Proof.* First we will prove the following restricted version of the result: if $f : \mathbb{R} \to \mathbb{R}$ is such that, for a given $n \in \mathbb{N}$, we have $\Delta_h^n f(x) = 0$, for all $x > 0$ and $h > 0$, then $f|_{\mathbb{R}^+}$ is a polynomial of order $n - 1$.

We will do so by induction on $n$:

The case $n = 1$ is obvious; $\Delta_h f(x) = 0$, that is $f(x + h) = f(x)$, for any $x, h > 0$, is the same as $f(y) = f(x)$, for any $y > x > 0$. In other words, $f$ is constant in $\mathbb{R}^+$.

Now assume the result holds for $n$ and that $\Delta_h^{n+1} f(x) = 0$, for all $x, h > 0$. Consider the function $p : \mathbb{R} \to \mathbb{R}$, defined by $p(x) = \dfrac{f(x) - f(0)}{x}$, when $x > 0$, and $p(x) = 0$, for $x \le 0$. Note that since $xp(x) = f(x) - f(0)$, for any $x \ge 0$, we have $0 = \Delta_h^{n+1}(f(x) - f(0)) = \Delta_h^{n+1}(xp(x))$, for all $x, h > 0$. Then, by the particular case of Leibniz rule provided by the upcoming identity (A.7) we get (for $x, h > 0$)

$$\begin{aligned}
0 = \Delta_h^{n+1}(xp(x)) &= x\Delta_h^{n+1} p(x) + (n + 1)h\Delta_h^n p(x + h) \\
&= x\left(\Delta_h^n p(x + h) - \Delta_h^n p(x)\right) + (n + 1)h\Delta_h^n p(x + h) \\
&= \left(x + (n + 1)h\right)\Delta_h^n p(x + h) - x\Delta_h^n p(x) ,
\end{aligned}$$

therefore, for $x, h > 0$,

$$\left(x + (n + 1)h\right)\Delta_h^n p(x + h) = x\Delta_h^n p(x) . \tag{A.2}$$

Unfortunately we cannot evaluate the previous identity directly on $x = 0$. However, if we recall the well known general identity

$$\Delta_h^n g(x) = \sum_{k=0}^n (-1)^{n-k}\binom{n}{k} g(x + kh) , \tag{A.3}$$

and the definition of $p$, we get, for $x, h > 0$,

$$x\Delta_h^n p(x) = \sum_{k=0}^{n} (-1)^{n-k}\binom{n}{k} x \frac{f(x+kh) - f(0)}{x+kh} , \tag{A.4}$$

which converges to zero, when $x \to 0$. Therefore, it follows from (A.2) and the continuity of $p$, in $\mathbb{R}^+$, that $\Delta_h^n p(h) = 0$, for $h > 0$. Considering $x = (k-1)h$ in (A.2), for $k \geq 2$, we conclude that $(n-k)h\Delta_h^n p(kh) = (k-1)h\Delta_h^n p((k-1)h)$. Inductively we determine that $\Delta_h^n p(kh) = 0$, for all $k \in \mathbb{N}$. We have thus concluded that

$$\Delta_h^n p(x) = 0, \text{ for all } x > 0 \text{ and all } h \in \left\{\frac{x}{k} : k \in \mathbb{N}\right\}.$$

Additionally, when $h = x/k$, it also holds that $h = (x+jh)/(k+j)$, implying that

$$\Delta_h^n p(x+jh) = 0, \text{ for all } x > 0 , \text{ all } h \in \left\{\frac{x}{k} : k \in \mathbb{N}\right\} \text{ and all } j \in \mathbb{N}_0 . \tag{A.5}$$

Moreover, given $m \in \mathbb{N}$, using (A.5) and the upcoming identity (A.6) we conclude that $\Delta_{mh}^n p(x) = 0$, provided $h = x/k$ and $m, k \in \mathbb{N}$, i.e.,

$$\Delta_h^n p(x) = 0, \text{ for for all } x > 0 \text{ and } h \in \{xQ : Q \in \mathbb{Q}^+\}.$$

By continuity of $p$, in $\mathbb{R}^+$, we finally know that

$$\Delta_h^n p(x) = 0, \text{ for all } x, h > 0 .$$

By the induction hypothesis, when restricted to $\mathbb{R}^+$, $p$ is a polynomial of order $n-1$ and, therefore, there exists a polynomial $q : \mathbb{R} \to \mathbb{R}$, of order $n$, such that

$$f(x) = q(x), \text{ for all } x > 0 .$$

If we now apply the general identity (A.3), to $f$, at the point $x = -h/2$, with $h > 0$, and take into consideration that, for $k \in \mathbb{N}_0$, $-h/2 + kh < 0 \Leftrightarrow k = 0$, we get

$$\Delta_h^{n+1} f(-h/2) = \sum_{k=0}^{n+1} (-1)^{n+1-k}\binom{n+1}{k} f(-h/2 + kh)$$

$$= f(-h/2) + \sum_{k=1}^{n+1} (-1)^{n+1-k}\binom{n+1}{k} q(-h/2 + kh)$$

$$= f(-h/2) - q(-h/2) + \sum_{k=0}^{n+1} (-1)^{n+1-k}\binom{n+1}{k} q(-h/2 + kh)$$

$$= f(-h/2) - q(-h/2) + \Delta_h^{n+1} q(-h/2) .$$

Since, for all $x$, $\Delta_h^{n+1} f(x) = 0$ by hypothesis, and $\Delta_h^{n+1} q(x) = 0$, because $q$ is a polynomial of order $n$, we conclude that $f(-h/2) = q(-h/2)$, for all $h > 0$. By continuity $f = q$, in the real line. $\square$

In the proof of the previous theorem we relied on:

**Lemma 3.** *Let $n, k \in \mathbb{N}$. There exist real coefficients $\{a_j^{(n)}\}_{j \in [k]}$ such that, for any function $p : \mathbb{R} \to \mathbb{R}$, the following identity holds*

$$\Delta_{kh}^n p(y) = \sum_{j=0}^{n(k-1)} a_j^{(n)} \Delta_h^n p(y+jh) . \tag{A.6}$$

*Proof.* If $n = 1$,

$$\Delta_{kh} p(y) = p(y+kh) - p(y) = \sum_{j=0}^{k-1} p\big(y + (j+1)h\big) - p(y+jh) = \sum_{j=0}^{k-1} \Delta_h p(y+jh).$$

In this case $a_j^{(1)} = 1$, for $j = 1, \ldots, k-1$. Assuming the result for $n$ we will prove it for $n+1$.

$$\Delta_{kh}^{n+1} p(y) = \Delta_{kh}\left(\Delta_{kh}^n p(y)\right) = \Delta_{kh}\left(\sum_{j=0}^{n(k-1)} a_j^{(n)} \Delta_h^n p(y+jh)\right) = \sum_{j=0}^{n(k-1)} a_j^{(n)} \Delta_{kh}\left(\Delta_h^n p(y+jh)\right)$$

Now, using the result for $n = 1$, that is $\Delta_{kh} p(y) = \sum_{i=0}^{k-1} \Delta_h p(y + ih)$, we get

$$
\begin{aligned}
\Delta_{kh}^{n+1} p(y) &= \sum_{j=0}^{n(k-1)} a_j^{(n)} \sum_{i=0}^{k-1} \Delta_h^{(n+1)} p(y + jh + ih) \\
&= \sum_{m=0}^{(n+1)(k-1)} \left( \sum_{\substack{i+j=m \\ 0 \le i \le k-1 \\ 0 \le j \le n(k-1)}} a_j^{(n)} \right) \Delta_h^{(k+1)} p(y + mh) \\
&= \sum_{m=0}^{(n+1)(k-1)} a_m^{(n+1)} \Delta_h^{(k+1)} p(y + mh) ,
\end{aligned}
$$

where the second equality arises from the change of variables $(i, j) \mapsto (m = i + j, j)$, and the last corresponds to the recursive definition of the coefficients $a_m^{(n+1)}$. $\qquad \square$

In the proof of the last theorem we also used the following special case of the well known Leinbiz rule for finite difference, the proof of which we present here for the sake of completeness.

**Lemma 4.** *For any function $g : \mathbb{R} \to \mathbb{R}$,*

$$
\Delta_h^{n+1}(xg(x)) = x\Delta_h^{n+1}(g(x)) + (n + 1)h\Delta_h^n(g(x + h)). \tag{A.7}
$$

*Proof.* For $n = 0$,

$$
\Delta_h(xg(x)) = (x + h)g(x + h) - xg(x) = x\Delta_h(g(x)) + hg(x + h).
$$

Assume the identity is valid for $n$. Then

$$
\begin{aligned}
\Delta_h^{n+1}(xg(x)) &= \Delta_h^n\left(\Delta_h(xg(x))\right) = \Delta_h^n\left((x + h)g(x + h)\right) - \Delta_h^n(xg(x)) \\
&= \left((x + h)\Delta_h^n g(x + h) + (nh)\Delta_h^{n-1} g(x + 2h)\right) - \left(x\Delta_h^n g(x) + (nh)\Delta_h^{n-1} g(x + h)\right) \\
&= x\Delta_h^{n+1} g(x) + h\Delta_h^n g(x + h) + (nh)\Delta_h^n g(x + h) \\
&= x\Delta_h^{n+1} g(x) + (n + 1)h\Delta_h^n g(x + h) .
\end{aligned}
$$

$\qquad \square$

For our next result we will also need a simple version of the chain rule for finite differences. To state it, we need to recall that given $g : \mathbb{R}^2 \to \mathbb{R}$ we can define the variations with respect to the second variable by

$$
\frac{\Delta_h g}{\delta y}(x, y) := g(x, y + h) - g(x, y) .
$$

It is then easy to see that, given $f : \mathbb{R} \to \mathbb{R}$ and $\alpha, \beta \in \mathbb{R}$, we have

$$
\frac{\Delta_h}{\delta y} \left[ f(\alpha x + \beta y) \right] = \left( \Delta_{\beta h} f \right)(\alpha x + \beta y) . \tag{A.8}
$$

We now have all we need to state and prove the final result of this paper:

**Theorem 7.** *Let $z = (z_i)$ and $w = (w_i)$ be totally non-aligned, meaning that*

$$
\begin{vmatrix} z_i & w_i \\ z_j & w_j \end{vmatrix} \ne 0 \ , \ \text{for all } i \ne j , \tag{A.9}
$$

*and let $\sigma : \mathbb{R} \to \mathbb{R}$. If there exists $u \in \mathbb{R}^N$, with all components non-vanishing, such that*

$$
\sum_{i=1}^N u_i \, \sigma(\theta_1 z_i + \theta_2 w_i) = 0 \ , \quad \text{for every } (\theta_1, \theta_2) \in \mathbb{R}^2 , \tag{A.10}
$$

*then $\Delta_{\boldsymbol{h}}^{N-1} \sigma(x) = 0$, for all $x \in \mathbb{R}$ and all $\boldsymbol{h} \in \mathbb{R}^{N-1}$.*

*Proof.* The totally non-aligned condition implies, in particular, that no more than one $z_i$ can vanish. Therefore, by rearranging the indices, we can guarantee that $z_i \neq 0$, for all $i \in [N - 1]$. Then, since $u_1 \neq 0$, we rewrite (A.10) as

$$\sigma(\theta_1 z_1 + \theta_2 w_1) = \sum_{i=2}^{N} u_i^{(1)} \sigma(\theta_1 z_i + \theta_2 w_i) , \quad \text{for every } (\theta_1, \theta_2) \in \mathbb{R}^2 , \tag{A.11}$$

where $u_i^{(1)} := -u_i/u_1 \neq 0$.

Next we consider the change of variables, $(\theta_1, \theta_2) \mapsto (x_1, y_1)$, defined by

$$\begin{cases} x_1 = z_1 \theta_1 + w_1 \theta_2 \\ y_1 = \theta_2 , \end{cases} \tag{A.12}$$

which is clearly a bijection since $z_1 \neq 0$. In the new variables we have

$$\theta_1 z_i + \theta_2 w_i = \frac{z_i}{z_1}(\theta_1 z_1 + \theta_2 w_1) + w_1 \left( \frac{w_i}{w_1} - \frac{z_i}{z_1} \right) \theta_2 = \alpha_1^i x_1 + \beta_1^i y_1 ,$$

where

$$\alpha_1^i := \frac{z_i}{z_1} \quad \text{and} \quad \beta_1^i := \frac{z_1 w_i - z_i w_1}{z_1} .$$

Applying these to (A.11) gives

$$\sigma(x_1) = \sum_{i=2}^{N} u_i^{(1)} \sigma(\alpha_1^i x_1 + \beta_1^i y_1) , \quad \text{for all } (x_1, y_1) \in \mathbb{R}^2 . \tag{A.13}$$

It turns out that by taking variations with respect to the second variable we can iterate this process. In fact, we will now prove that, for all $0 \leq k \leq N - 2$ and all non-vanishing $h_i$, $i \in [N - 1]$, the following recursive identity holds:

$$\left( \Delta_{\boldsymbol{h}_k^{k+1}}^k \sigma \right)(x_{k+1}) = \sum_{i=k+2}^{N} u_i^{(k+1)} \left( \Delta_{\boldsymbol{h}_k^i}^k \sigma \right)(\alpha_{k+1}^i x_{k+1} + \beta_{k+1}^i y_{k+1}) , \quad \text{for all } (x_{k+1}, y_{k+1}) \in \mathbb{R}^2 , \tag{A.14}$$

where the coefficients are determined by

$$u_i^{(k+1)} = -u_i^{(k)}/u_{k+1}^{(k)} \neq 0 , \tag{A.15}$$

$$\alpha_j^i = \frac{z_i}{z_j} \quad \text{and} \quad \beta_j^i = \frac{z_j w_i - z_i w_j}{z_j} , \tag{A.16}$$

the change of variables is defined by

$$\begin{cases} x_{k+1} = \alpha_k^{k+1} x_k + \beta_k^{k+1} y_k \\ y_{k+1} = y_k , \end{cases} \tag{A.17}$$

and the increment vectors are set according to

$$\boldsymbol{h}_k^i = \left( \beta_k^i h_k, \beta_{k-1}^i h_{k-1}, \dots, \beta_1^i h_1 \right) . \tag{A.18}$$

Notice that all components of $\boldsymbol{h}_k^i$ are non-vanishing.

The proof follows by induction: The $k = 0$ case corresponds to (A.13). So let us assume that the identity holds for $0 \leq k \leq N - 3$. Then, by taking variations of (A.14) with respect to $y_{k+1}$ and increment $h = h_{k+1} \neq 0$, we can use the chain rule (A.8) to obtain

$$0 = \sum_{i=k+2}^{N} u_i^{(k+1)} \left( \Delta_{\boldsymbol{h}_{k+1}^i}^{k+1} \sigma \right)(\alpha_{k+1}^i x_{k+1} + \beta_{k+1}^i y_{k+1}) , \quad \text{for all } (x_{k+1}, y_{k+1}) \in \mathbb{R}^2 , \tag{A.19}$$

where $\boldsymbol{h}_{k+1}^i = (\beta_{k+1}^i h_{k+1}, \boldsymbol{h}_k^i)$, which can be rewritten as

$$\left( \Delta_{\boldsymbol{h}_{k+1}^{k+2}}^{k+1} \sigma \right)(\alpha_{k+1}^{k+2} x_{k+1} + \beta_{k+1}^{k+2} y_{k+1}) = \sum_{i=k+3}^{N} u_i^{(k+2)} \left( \Delta_{\boldsymbol{h}_k^i}^{k+1} \sigma \right)(\alpha_{k+1}^i x_{k+1} + \beta_{k+1}^i y_{k+1}) , \tag{A.20}$$

with $u_i^{(k+2)} = -u_i^{(k+1)}/u_{k+2}^{(k+1)} \neq 0$.

Following the iterative procedure, consider the change of variables $(x_{k+1}, y_{k+1}) \mapsto (x_{k+2}, y_{k+2})$ defined by

$$\begin{cases} x_{k+2} = \alpha_{k+1}^{k+2} x_{k+1} + \beta_{k+1}^{k+2} y_{k+1} \\ y_{k+2} = y_{k+1} \,, \end{cases} \tag{A.21}$$

and observe that it is a bijection, since $k + 2 \leq N - 1$ implies that $z_{k+2} \neq 0 \Leftrightarrow \alpha_{k+1}^{k+2} \neq 0$. In these new variables

$$\alpha_{k+1}^i x_{k+1} + \beta_{k+1}^i y_{k+1} = \frac{\alpha_{k+1}^i}{\alpha_{k+1}^{k+2}} \left( \alpha_{k+1}^{k+2} x_{k+1} + \beta_{k+1}^{k+2} y_{k+1} \right) + \left( \beta_{k+1}^i - \beta_{k+1}^{k+2} \frac{\alpha_{k+1}^i}{\alpha_{k+1}^{k+2}} \right) y_{k+1}$$

$$= \alpha_{k+2}^i x_{k+2} + \beta_{k+2}^i y_{k+2} \,,$$

with the last identity requiring some algebraic manipulations to be established. So we see that, in the new variables, (A.20) becomes

$$\left( \Delta_{\boldsymbol{h}_{k+1}^{k+2}}^{k+1} \sigma \right) (x_{k+2}) = \sum_{i=k+3}^{N} u_i^{(k+2)} \left( \Delta_{\boldsymbol{h}_k^i}^{k+1} \sigma \right) (\alpha_{k+2}^i x_{k+2} + \beta_{k+2}^i y_{k+2}) \,, \quad \text{for all } (x_{k+2}, y_{k+2}) \in \mathbb{R}^2 \,, \quad \text{(A.22)}$$

as desired. This closes the induction proof that establishes the validity of (A.14) for all $0 \leq k \leq N-2$. By choosing $k = N - 2$ in that identity we obtain

$$\left( \Delta_{\boldsymbol{h}_{N-2}^N}^{N-2} \sigma \right) (x_{N-1}) = u_N^{(N)} \left( \Delta_{\boldsymbol{h}_{N-2}^N}^{N-2} \sigma \right) (\alpha_{N-1}^N x_{N-1} + \beta_{N-1}^N y_{N-1}) \,, \quad \text{for all } (x_{N-1}, y_{N-1}) \in \mathbb{R}^2 \,. \quad \text{(A.23)}$$

Finally, if we take variations of the last equation with respect to $y_{N-1}$ and increment $h = h_{N-1} \neq 0$, and recall that $u_N^{(N)} \neq 0$, we arrive at

$$0 = \left( \Delta_{\boldsymbol{h}_{N-1}^N}^{N-1} \sigma \right) (\alpha_{N-1}^N x_{N-1} + \beta_{N-1}^N y_{N-1}) \,, \quad \text{for all } (x_{N-1}, y_{N-1}) \in \mathbb{R}^2 \,, \quad \text{(A.24)}$$

where $\boldsymbol{h}_{N-1}^N = (\beta_{N-1}^N h_{N-1}, \boldsymbol{h}_{N-2}^N)$. Since, in view of (A.9), all $\beta_j^i$, with $i \neq j$, are non-vanishing we conclude that $\left( \Delta_{\boldsymbol{h}}^{N-1} \sigma \right) (x) = 0$, for all $x \in \mathbb{R}$ and all $\boldsymbol{h} \in \mathbb{R}^{N-1}$. $\qquad \square$

