# OpenReview forum: "On the Positive Definiteness of the Neural Tangent Kernel"
_ICLR.cc/2024/Conference — Submitted to ICLR 2024_

### Official Review · Reviewer_vtuM · 2023-10-18

**Soundness:** 3 good
**Presentation:** 3 good
**Contribution:** 2 fair
**Rating:** 3
**Confidence:** 3

**Summary:**

This paper studies the minimum eigenvalue of the neural tangent kernel (NTK), which is an essential problem for analyzing the convergence and generalization of over-parameterized neural networks. They have two main results. First, for a multi-layer network with activated biases and a continuous, differentiable, and non-polynomial activation function, the NTK is positive definite. Second, for a multi-layer network without bias and a continuous, differentiable, and non-polynomial activation function, if the training data points are pairwise non-proportional, then the NTK is positive definite.

**Strengths:**

The paper's results improve on previous results in two ways: first, they apply to more general activation functions; second, they do not require strong assumptions on the training data. The paper is well-written, with clear statements of the theorems and rigorous proofs.

**Weaknesses:**

My main concern with this paper is the usefulness of its results. First, the two results only show that the minimum eigenvalue of the kernel is non-zero. However, to analyze the convergence rate of over-parameterized neural networks, we need an explicit bound on the minimum eigenvalue in terms of the network parameters. Therefore, by combining the results with convergence theory, we can only deduce that gradient descent on those neural networks will minimize the training loss to zero. However, we still do not know the training costs. Second, in recent years, many works have pointed out the limitations of NTK and infinite-width neural networks. To apply the NTK theory, we may assume that the width should be $m=\Omega(n^4)$, which is too impractical. Therefore, this paper may only have a limited broader impact on the deep learning theory community.

**Questions:**

Some typos:

P3: “As mentioned above We generalize”: “We”->”we”

P7, $K_X^{(2)}$: an extra “)”

---

> ### Author Response · Authors · 2023-11-20
>
> ## Reply to "Weaknesses":
>
> It is true that knowing an explicit lower bound of the first eigenvalue of the NTK would be more useful in practice. However, there has been recent substancial interest in simply establishing the positivity of the NTK and up to the present paper no fully general (sharp) result was known. For example, previous works were unable to completely eradicate unnecessary and quite restrictive hypothesis such as that the data points lie in a sphere (which is never the case in practice) or that the activation function is analytic (which for example fails for the ReLu). Our paper completely solves the question of establishing the positivity of the NTK by making no unnecessary hypothesis and proving such positivity in full generality.
>
> It is also true, as the reviewer points out, that there are several shortcomings of considering infinitely wide networks in which the NTK is constant. This is therefore a valid critique, but in our defense, the NTK continues to be a fundamental tool in understanding wide neural networks.
>
>
> ## Reply to "Questions":
>
> Thank you for pointing out these typos. We have made these changes in the text.

---

### Official Review · Reviewer_jE8k · 2023-10-26

**Soundness:** 3 good
**Presentation:** 3 good
**Contribution:** 1 poor
**Rating:** 5
**Confidence:** 3

**Summary:**

This paper analyzes the neural tangent kernel at the infinite width limit. It shows that NTK is strictly positive definite, as long as the activation function is not a polynomial and data is non-degenerate (pairwise non-proportional, if no bias). The major technique it uses is Theorem 3, about a characterization of polynomial functions.

**Strengths:**

The main results of this paper require milder assumptions than prior works. Particularly, it does not require the unit sphere data assumption. Compared to Du et. al. 2019, it also does not require the activation function to be analytic.

The paper is clearly written. Main techniques are highlighted, so that intuitions can be easily seen.

**Weaknesses:**

My concern is on the significance of the results and the technical novelty.
Similar results/claims already exist with a little bit stronger assumptions. For example, Du et. al. 2019 showed the same thing, just additionally required unit sphere data, and analytic activation functions. I am afraid this improvement in this paper is not enough to meet the ICLR acceptance standard.

In addition, most parts of the proofs (except the application of Theorem 3) in the main content are common treatments which can be found easily in literature. It seems a bit tedious for those who are familiar with the topic. Theorem 3 seems a bit novel (at least to my knowledge), but not technically hard to prove. Hence, I also have concern on the significance of technical novelty.

**Questions:**

no further questions

---

> ### Author Response · Authors · 2023-11-20
>
> ## Reply to "Weaknesses":
>
> Our article uses techniques which substantially deviate from those used to analyse the positivity of the NTK in the existing literature. For instance, we rely in a new characterization of polynomials using finite differences. It is this technical novelties that allows to prove a sharp result finally establishing the positivity of the NTK in full generality. In comparison the mentioned paper by Du et al. makes very strong assumption on:
>
> (i) The distribution of the dataset which is assumed to lie in a sphere. This is never the case in practice.
> (ii) The analyticity of the activation function. This leaves out widely used activation functions such as the ReLu.
>
> In addition to improving the results, we substantially simplify the proofs by not relying on back-boxes involving for instance technical results on Hermite polynomials which only apply to solve the problem when the data is distributed in a sphere.
>
>
>
> ## Reply to "Questions":
>
> There were no questions.

---

### Official Review · Reviewer_onui · 2023-10-29

**Soundness:** 3 good
**Presentation:** 3 good
**Contribution:** 3 good
**Rating:** 6
**Confidence:** 3

**Summary:**

The paper is dedicated to sufficient conditions for the positive definiteness of NTK. It is proved that the architecture with a bias term and a non-polynomial activation function automatically leads to positive-definite NTK. The proof is based on Theorem 3, which states that functions sigma(a[i]x+b[i]y), i=1,...,n are linearly independent if [a[i],b[i]] is not a multiple of [a[j],b[j]] for all i,j and sigma is not polynomial. An easy proof of Theorem 3 is given for the case when sigma is many times differentiable. In the appendix, a more elaborate proof is given for a general case. To avoid differentiability finite differences are analyzed instead. Then a case of an architecture with only one hidden layer becomes quite straightforward. A general case is treated in Proposition 1, in which it is proved that positive definiteness of NTK for lower layers inductively guarantees positive definiteness of NTK for the next layer.

Major claims seem correct, proofs are convincing. The paper is purely theoretical. A major weakness is a lack of deeper discussions about what these results give us for a better understanding of NNs.

Minor correction on page 1: as emerged -> has emerged

**Strengths:**

Mathematically clean, at least from the first site I could not find any issues.

**Weaknesses:**

There is no any discussion of proved results in the context of NTL theory. The fact that positive definiteness is somehow related to memorization is only mentioned. Also, experimental part is absent.

**Questions:**

Non-polynomiality of activation function also plays a key role in Universal approximation theorem as was probed by Moshe Leshno et al in 1993 and later Allan Pinkus in 1999. So a natural question is how it is related to the proved property that non-polynomiality leads to positive definiteness of NTK?

---

> ### Author Response · Authors · 2023-11-20
>
> ## Reply to "Weaknesses":
>
> We have an introduction which though brief does mention the important practical aspects of needing to have a positive definite NTK. We also review the existing literature in section 1.3 and we have done our best to summarize it.
>
>
> ## Reply to "Questions":
>
> This is a very good and interesting question. The "universal approximation theorem" and "positive definiteness of the NTK" are intimately linked. Our result on the non-polinomiality indeed links the two, even if indirectly.
>
> More directly, one can naively understand that there must be a relation because the positivity of the NTK (for such generic data sets) is related to the ability of the network to memorize such a training set (with zero error in the infinite width case). On the other hand, being able to memorize any such generic data sets implies the network has enough expressivity to approximate a large class of functions.
>
> Alternatively, there are more geometric explanations. We shall give here one such in a vague form. Neural networks can be interpreted as maps from the spaces of their parameters to a function space, the NTK corresponding to the pushforward of the inverse (flat) metric on the space of parameters to its image. The "universal approximation theorem" applies when this map has a dense image. As for the "positivity of the NTK", it applies when evaluating the networks at points (the data points) corresponding to independent coordinates in the image. Therefore, having an upper bound on the rank of the NTK implies having an upper bound on the dimension of the image, and therefore a failure of the universal approximation theorem.
>
> If the reviewer finds it needed, we are happy to incorporate this explanation in the body of the text. We had originally omitted them due to space restrictions.

---

### Official Review · Reviewer_BSSD · 2023-10-31

**Soundness:** 3 good
**Presentation:** 2 fair
**Contribution:** 1 poor
**Rating:** 3
**Confidence:** 4

**Summary:**

This paper analyzes the positive definiteness of the neural tangent kernel, which is the inner product of the gradient of the network function w.r.t. the weights in the infinite limit of the width. Compared to the previous work (Du et al. 2019), this paper shows that the activation function does not need to be analytic but being continuous and differentiable a.e. is sufficient to prove the result.

**Strengths:**

This paper gives a nice introduction to the background of the neural tangent kernel and its importance. The proof is well presented and is easy to follow.

**Weaknesses:**

1. The contribution made in this paper is incremental.  Compared to the previous work (Du et al. 2019), it only improves the condition a little. Besides, I agree proving the positive definiteness of the NTK is an interesting question but I don't think it's significant enough for ICLR.

2. Theorem 1 is wrong. If two data samples are the same, then the NTK is not positive definite. I think the authors missed the condition that data samples are different.

3. I don't understand the point of section 2.1. It spends two pages explaining the results that do not satisfy the condition of Theorem 1 ( the activation function is assumed to be $C^{N-1}$ in this section). Besides, the paragraph is repeated at the end of page 5.

4. In the proof of Theorem 4, it says ''with the $z_i$ pairwise distinct and $\beta \neq 0$, in the view of Theorem 3...". However, Theorem 3 requires totally non-aligned where being pairwise distinct is not sufficient.

5. I don't think whether there is a bias or not is an important thing. With bias, x can be viewed as $[x,1]$. Therefore, x being pairwise non-proportional becomes [x,1] being pairwise distinct.

Given the weaknesses I have listed, I believe this paper needs some major revision.

**Questions:**

I didn't follow the proof of Proposition 1. ''under the previous circumstances, $\sigma$ must be a constant. I am unclear about the mentioned circumstances and how to see that $\sigma$ is a constant.

---

> ### Author Response · Authors · 2023-11-20
>
> ## Reply to "Weaknesses"
>
> (1) There have been several works which have attempted to establish the positivity of the NTK. However, none have so far been able to completely eradicate unnecessary and quite restrictive hypothesis such as that the data points lie in a sphere (which is never the case in practice) or that the activation function is analytic (which for example fails for the ReLu). Our paper completely solves the question of establishing the positivity of the NTK by making no unnecessary hypothesis and proving such positivity in full generality. %Without this contribution the literature would be incomplete and at an unsatisfactory stage.
>
> (2) It was stated in the preamble to Theorem 1, when defining positive definite Kernel, that we will always consider a data set without repeated points. To further emphasize it, we have now abstracted this into Definition 1 and make reference to it in the statement of Theorem 1.
>
> (3) Given that our technique is novel in this area, and the full proof is substantially involved occupying several pages in the appendix, we decided it would be instructive and clarifying to have a simpler case solved in full detail in the body the article.
>
>  (4) In the scope of Theorem 4, meaning with biases (equivalently $\beta \neq 0$) the $z_i$ being all distinct is equivalent to $z=(z_1,\ldots , z_N)$ being totally non-aligned with the bias vector (as was stated in Lemma 1). Hence, we can invoke Theorem 3 at this stage.
>
>  (5) In fact, the reviewer mentions this same fact in the upcoming point 5 of the weaknesses section.
> To further elucidate this point, we moved this observation from Lemma 1 to the relevant point during the proof of Theorem 4.
>
> (6) Indeed, this is true and we have added a remark after the statement of Theorem 2 concerning that point. However, for the sake of stating the Theorems in a way that can be easily understood by casual readers, it is important to highlight the different necessary conditions on the data set to ensure the positiveness of the NTK.
>
> ## Reply to "Questions"
>
> The words "under the previous circumstances" simply refer to the hypothesis of the Theorem which have already been used. At this stage, the fact that $\sigma$ is constant follows immediately from the formula $\sigma(x)=-\sigma(0) \sum_{i=1}^{N-1}\frac{u_i}{u_N}$ written in the line immediately above (in the paper).

---

### Author Response · Authors · 2023-11-20

We thank the reviewers for their time in revising our article. We have submitted responses specifically addressing the points raised by the referees.


## Changes made:


(1) We have introduced Definition 1 and make reference to it in Theorem 1. This makes it more clear that no data points are the same.

(2) We have added a remark after Theorem 2 to help clarify the difference in the assumption at the level of the training set between Theorem 1 and Theorem 2.

(3) We added the explanation that the $z_i$ being all distinct is equivalent to $z=(z_1,\ldots , z_N)$ being totally non-aligned with the bias vector in the proof of Theorem 4. This was previously observed in Lemma 1, but following a reviewer's question we moved it to the proof of Theorem 4.

(4) We corrected some typos.

---

### Meta-Review · Area_Chair_BXry · 2023-12-02

**Metareview:**

The paper aims to establish sufficient conditions for ensuring the positive-definiteness of the Neural Tangent Kernel and shows that neural network architectures incorporating bias terms and non-polynomial activation functions exhibit this property. While the paper contains somewhat interesting results, the general impression is that the contributions made, relative to prior works, are somewhat incremental, and the significance of the results appears to be limited.

**Justification For Why Not Higher Score:**

The contributions of the paper, relative to prior works, are somewhat incremental, and the significance of the results appears to be limited.

**Justification For Why Not Lower Score:**

N/A

---

### Decision · Program_Chairs · 2024-01-16

Reject